# Extracurricular Functions of tRNA Modifications in Microorganisms

**DOI:** 10.3390/genes11080907

**Published:** 2020-08-07

**Authors:** Ashley M. Edwards, Maame A. Addo, Patricia C. Dos Santos

**Affiliations:** Department of Chemistry, Wake Forest University, Winston-Salem, NC, 27101, USA; edwaam16@wfu.edu (A.M.E.); addoma16@wfu.edu (M.A.A.)

**Keywords:** tRNA modification, 2-thiouridine, 4-thiouridine, queuosine, 2-methylthio-*N*^6^-(cis-hydroxyisopentenyl) adenosine, uridine 5-oxyacetic acid, 2-thioribothymidine, epitranscriptome

## Abstract

Transfer RNAs (tRNAs) are essential adaptors that mediate translation of the genetic code. These molecules undergo a variety of post-transcriptional modifications, which expand their chemical reactivity while influencing their structure, stability, and functionality. Chemical modifications to tRNA ensure translational competency and promote cellular viability. Hence, the placement and prevalence of tRNA modifications affects the efficiency of aminoacyl tRNA synthetase (aaRS) reactions, interactions with the ribosome, and transient pairing with messenger RNA (mRNA). The synthesis and abundance of tRNA modifications respond directly and indirectly to a range of environmental and nutritional factors involved in the maintenance of metabolic homeostasis. The dynamic landscape of the tRNA epitranscriptome suggests a role for tRNA modifications as markers of cellular status and regulators of translational capacity. This review discusses the non-canonical roles that tRNA modifications play in central metabolic processes and how their levels are modulated in response to a range of cellular demands.

## 1. Introduction

Post-transcriptional modifications expand the role of ribonucleic acids (RNAs) by introducing additional chemical functionalities to their associated nucleotides. Thus far, over 150 modifications have been discovered in RNA, 93 of which are exclusively found in transfer RNAs (tRNAs) [1]. A large number of these modifications have been characterized and range from simple methylations to more complex hypermodified species such as wybutosine (yW) and queuosine (Q) [2,3,4,5]. Modifications are present throughout the tRNA structure (Figure 1) and their functions can be predicted based on their locations within the molecule [6]. The presence of these modifications, along with the widely conserved 3′-CCA sequence used for aminoacylation, serve as prerequisites for fully functional tRNAs used during protein synthesis [7]. Specific modifications found within the anticodon loop influence translational accuracy and/or efficiency [8]. Modifications located in the acceptor stem, the D-loop, and the TΨC-loop are often involved in structural stabilization and can also serve as identification elements during RNA–protein recognition events [6,9,10].

One way modifications in tRNA guarantee fidelity of translation is by influencing codon discrimination. This is important in situations when two amino acids are defined by codons that vary by only one nucleotide, such as methionine (AUG) and isoleucine (AUA/U/C). In bacteria, accurate recognition of the AUA codon requires tRNA^Ile2^ to have 2-lysidine (k^2^C) at position 34 to prevent misacylation by methionyl-tRNA synthetase (MetRS) [11]. Additionally, the N4-acetylcytidine (ac^4^C) modification at position 34 of tRNA^Met^ acts as a positive identity determinant for MetRS [12]. Wobble base modifications also tune the reactivity of cognate tRNA substrates in aminoacyl tRNA synthetase (aaRS) reactions. Specifically, G34 and k^2^C constitute recognition elements for IleRS to discriminately aminoacylate tRNA^Ile1^ and tRNA^Ile2^, respectively [13,14]. In the archaea *Thermus thermophilus*, agmatidine (agm^2^C34) is found at the wobble position of tRNA^Ile2^ and similarly decodes the minor AUA isoleucine codon, while directly interacting with 70S ribosomal RNA (rRNA) [15].

The presence and conservation of selected modified bases serves as an identity determinant for aaRSs (Figure 2). 2-thiouridine (s^2^U), another wobble modification, is required for efficient charging of tRNA^Gln^ and it affects interactions with elongation factor-tu (Ef-Tu) [16]. In *Escherichia coli*, the presence of base-stacked U35/U36 found perpendicular to and in tandem with 5-methylaminomethyl-2-thiouridine (mnm^5^s^2^U34) in tRNA^Lys^ is essential for LysRS recognition and binding [17]. Likewise, 2-methyladenosine (m^2^A37) and pseudouridine (ψ38) in tRNA^Gln^ serve as stabilizing agents for anticodon loop nucleotides and are necessary for efficient GlnRS aminoacylation [18]. 5-methyluridine (m^5^U54) is necessary for tRNA interactions with Ef-Tu in the archaea *Thermus aquaticus* by altering their binding interface contacts and energies [19]. Structural changes resulting from photochemical reactions of 4-thiouridine (s^4^U8) impairs tRNA interactions with PheRS and ProRS [20]. These and other additional examples provided in this review show that the prevalence of these modifications not only impacts the tRNA’s ability to interact with their corresponding aaRSs, but also with essential partners including tRNA-modifying enzymes, Ef-Tu, ribosomes and coding RNA sequences (Figure 2) [17,19,21].

While the canonical functions of tRNA modifications in translation are well established, relatively little is known about how these modifications regulate and promote metabolic flow. Many organisms modulate their protein synthesis capacity during nutrient limitation by repressing cell growth until nutrient replete conditions are reestablished. For instance, amino acid starvation leads to the accumulation of uncharged tRNAs, resulting in the stringent response and reprogramming of metabolic processes [22]. Microbes employ a series of rapid adaptive responses when faced with nutritional and environmental stressors including changes in growth temperatures and pH, oxidative stress and hypoxia, and the presence of toxic compounds like antibiotics [23]. These stress factors cause reprogramming of tRNA modifications and tRNA pool composition, and lead to reduced growth rate and upregulation of cellular stress response elements [24]. Hence, the remodeling of the tRNA epitranscriptome contributes to adaptive responses enabling survivability during prolonged harsh growth conditions, and are considered evolutionary defense mechanisms [25].

Extracurricular regulatory functions of tRNA modifications are attributed to the impact that their presence or absence has on translational apparatus efficiency. The complex and emerging roles that tRNA modifications play in central metabolic control during stress-induced growth conditions have become one of the crucial cruxes of metabolomic and epitranscriptomic research. This review provides a survey of nutritional and environmental triggers that impact tRNA modification levels in microorganisms, while discussing the impact that altered levels of these modifications have on bacterial physiology. These non-canonical functionalities can be exploited in the characterization of evolutionary determinants of bacterial fitness and adaptation across diverse habitats. The dynamic nature of tRNA modifications expands their functions as regulatory agents, rather than solely acting as structural entities during translational processes [9,26,27].

## 2. Modifications Impacting tRNA Degradation

Nutritional and environmental stressors induce changes to the overall tRNA epitranscriptome, affecting the levels of fully functional and intact tRNA molecules (Figure 3). During amino acid starvation in the ciliate *Tetrahymena thermophila*, tRNA degradation processes selectively target deacylated or hypomodified tRNAs for destruction, resulting in the production of “tRNA halves” [28]. This process is thought to eliminate the improper use of those surplus molecules when metabolism demands a shift towards the stringent response. In bacteria, the presence of modified nucleosides affects the recognition and binding of RNA-specific endonucleases that produce these tRNA halves. For example, the binding specificities of *E. coli* tRNA^Lys^-specific anticodon nuclease (ACNase) are differentially regulated by the presence or absence of base modifications located in the tRNA anticodon loop [29]. However, it should be noted that tRNAs are modified at numerous bases and that these modifications are either simultaneously or sequentially installed onto tRNA. Thus, the cross-dependency of these tRNA modification processes obfuscates the establishment of additional functions given to any singular modification. These intricacies pose significant experimental challenges when assigning their individual roles in other biochemical and genetic controls.

tRNA quality control checkpoints play a pivotal role in understanding how organisms maintain fitness during resource limitation, suboptimal growth conditions, and pathogenesis. In *E. coli*, Q-modified tRNA^Tyr^, tRNA^His^, tRNA^Asn^, and tRNA^Asp^ are targeted for cleavage by the plasmid-encoded toxin ribonuclease, Colicin E5. This cleavage results in the stalling of protein synthesis [30]. Similarly, in yeast, the absence of the 7-methylguanosine (m^7^G46) modification initiates a rapid and specific degradation/de-acylation of tRNA^Val(AAC)^ substrates, culminating in a severe temperature sensitivity and slow growth phenotype that can only be reversed upon repletion of the m^7^G-modifying enzymes, Trm8p/Trm82p. Genetic complementation experiments demonstrated that this phenotype was, in fact, associated with the absence of tRNA^Val(AAC)^ due to hypomethylation-specific degradation [31]. m^7^G plays a structural role in stabilizing intramolecular interactions with residues 13 and 22 in the D-arm of certain tRNAs and its absence leads to thermodynamic destabilization, making the tRNA a prime target for tRNA-degrading machinery [32,33]. The m^7^G modification also occurs quite frequently in bacterial systems, but its effects on tRNA pool composition and degradation have yet to be explored. It is worth mentioning that in *E. coli*, the m^7^G-modifying enzyme YggH complements a *trm8-*Δ*/trm82*-Δ double mutant strain of *Saccharomyces cerevisiae* and fully rescues m^7^G biosynthesis [34]. Based on functional similarity, it is a reasonable proposal that bacterial systems also employ such degradation strategies as a tRNA quality control measure.

Another modification implicated in tRNA structural integrity is the highly conserved s^4^U8 modification found within the inner core of the D-stem loop. This modification is absent in Eukarya, but found in high abundance in Archaea and Bacteria domains. In *Vibrio cholerae*, hypothiolation of C4 at the U8 position allows targeted recognition by the RNA degradosome. Thus, the absence of this modification is linked to the inherent structural instability of tRNA in late growth phases. Interestingly, s^4^U-deficient aminoacylated tRNAs, when bound to Ef-Tu, exhibit significantly less RNA degradosome-mediated decay. This indicates that Ef-Tu may offer a protective role by competing with RNA degradation machinery for the binding of charged/hypomodified tRNAs [35]. This conclusion is supported by the fact that aminoacylation levels remain consistent across s^4^U-modified and hypomodified tRNAs, while rates of RNA degradation decrease based on the presence and binding of Ef-Tu to tRNAs. The idea that translational components such as Ef-Tu are able to exert shielding effects over charged tRNA substrates suggests their active participation in tRNA quality control [35]. Collectively, mechanisms targeting tRNA degradation demonstrate how non-essential tRNA modifications can have global metabolic consequences, particularly when organisms are subjected to exogenous stressors and limited nutrient pools.

## 3. Stress Factors Modulating tRNA Modifications

### 3.1. Nutritional Factors

Post-transcriptional tRNA modifications found in the anticodon loop are able to dynamically respond to shifts in nutrient availability. The wobble U34 position of certain tRNAs undergo extensive hypermodifications that stabilize codon–anticodon recognition. Modifications at this position ensure translational fidelity while optimizing protein synthesis. Specifically, the s^2^U modification of tRNA^Gln^, tRNA^Glu^ and tRNA^Lys^ is almost always further modified to 5-methyl-2-thiouridine (xm^5^s^2^U) derivatives through distinct pathways across diverse groups of bacteria [36]. Substitutions at the 5 and 2 positions of U34 induce puckering of the ribose moiety in the 3′- *endo* form, eliminating any steric clash that would be imposed on the tRNA by its bulky 2-thiocarbonyl group. U34 derivatives are positive determinants for reactivity with their corresponding aaRSs. In *E. coli*, the absence of mnm^5^ s^2^U decreases the binding capacity of tRNA^Glu^ to GluRS by nearly 100-fold [37], highlighting the importance of anticodon loop modifications in aaRS-substrate recognition as well as protein synthesis efficiency.

A regulatory role of s^2^U in tRNA has been proposed because the levels of this modification change in response to nutritional status. In budding yeast, the relative abundance of s^2^U fluctuates along with the availability of sulfur-containing amino acids. Under high levels of Cys and Met, ample levels of s^2^U signals accompanied increased translation and growth. However, when these sulfur sources become depleted, low levels of s^2^U initiates the salvage and synthesis of sulfur-containing and essential amino acids, such as Cys, Met, and Lys [38]. Furthermore, genes encoding proteins involved in carbohydrate metabolism are enriched in codons preferentially read by thiolated tRNAs. More recently, Laxman et al. also showed that U34 thiolation deficiency shifts metabolism away from nucleotide anabolism and shunts resources towards carbon storage and trehalose synthesis [39]. This metabolic response releases phosphate as a byproduct and reestablishes cellular phosphate equilibrium. These phenomena provide the experimental basis for a regulatory model in which thiolated tRNA^Gln,Glu,Lys^ act as sensors of sulfur-containing amino acid availability and as regulators of central metabolism. Thus, the observed fluctuations in U34 thiolation makes this modification one of many metabolic markers of “starvation state” or “growth state”. Regardless of their role as sensors and/or markers, these findings illustrate the importance of modified sulfur-containing tRNAs (s-tRNAs) in modulating translational competence in response to nutritional demands.

The sensitivity of anticodon loop modifications to cellular nutritional status is not a phenomenon exclusive to s-tRNAs and their s-modifying biosynthetic enzymes. In the fission yeast *Schizosaccharomyces pombe*, the 7-deaza-guanosine Q-modification is the result of an irreversible queuine-inserting, enzyme-catalyzed (eTGT) replacement of wobble G34 with the micronutrient queuine [40]. Eukaryotes are unable to *de novo* synthesize queuine and therefore rely on gut microflora and dietary supplementation for this process. Müller et al. found that in *S. pombe*, the presence of Q is necessary for the Pmt1-dependent methylation of C38 (m^5^C) located in the anticodon loop of tRNA^Asp^. During growth in queuine-supplemented medium, the in vivo stimulation of Pmt1-methylation on C38 was strikingly evident, and, when eTGT is removed, the increase in methylation was lost as well [41]. The dependency of m^5^C on the queuosinylation of G34 points to the sequential nature of two completely separate base modifications. This newfound reliance of C38 methylation on the presence of Q34 also provides another checkpoint that allows for the queuine-salvaging microflora to have a regulatory role over the host’s tRNA-modifying machinery. These findings imbue Q as a positive determinant for m^5^C synthesis and establish a new and indirect function for modulating the speed and precision of translational events at an interspecies level [42]. The intersecting nature of these pathways not only further iterates the complex roles of tRNA modifications in translation, but also gives insight into signaling networks of host-microbe interactions.

A37 directly adjacent to the 3′ end of the anticodon is another frequently modified position in the anticodon loop. Its conversion into 2-methylthio-*N*^6^-isopentenyl adenosine (ms^2^i^6^A) prevents aberrant intramolecular hydrogen bonding within the anticodon loop and ensures stability of the open loop structure required for correct base pairing [8,43]. Simple base alterations such as the addition of an isopentenyl group to adenosine (i^6^A) are among some of the most conserved modifications. Interestingly, ms^2^i^6^A responds to fluctuations in essential metabolite pools and varies with growth stage. In *Bacillus subtilis,* nutrient depletion initiates sporulation, which is accompanied by an increase in ms^2^i^6^A levels. During this stage, carbon starved cells transitioning from the exponential to stationary growth phase display a vast decrease in glycolytic metabolites and increased levels of pathway end-products like phosphoenolpyruvate (PEP), signaling glucose deprivation. Phosphate limitation causes a sharp decrease in phosphorylated glycolytic intermediates and also displays increased levels of tRNA thiomethylation [44]. The link between higher levels of ms^2^i^6^A under phosphate limitation supports the role of this modification as a marker for decreased cellular replication and nutrient limitation. Although this proposal awaits experimental validation, it is thought that accumulation of ms^2^i^6^A in late growth phases allows for enhanced ribosome–codon interactions with methylthiolated tRNA^Tyr^, and that these interactions may selectively translate sporulation-specific messenger-RNAs (mRNAs) [45]. The coordination of these physiological and structural changes suggests that ms^2^i^6^A functions either as a synchronous marker or as a sensor of nutrient starvation in *B. subtilis* and possibly in other spore-forming bacterial species.

Iron availability also impacts modifications to A37. The tRNA-modifying Fe–S dependent-enzyme, MiaB, is highly sensitive to intracellular iron metabolism and oxidative stress [46]. In *E. coli*, these conditions also indirectly impact the expression of Fur, the master negative regulator of iron homeostasis. In this bacterium, the translation of Fur is tightly coupled to the synthesis of the Uof, a leader peptide whose coding sequence is located upstream of *fur*. The Uof coding sequence includes a Ser–UCA codon at position 6 of the *uof* gene that is preferentially read by methylthiolated tRNA^Ser(UGA)^, linking iron homeostasis to ms^2^i^6^A production [47,48]. This example highlights the importance of tRNA modifications in one of many biological feedback loops that regulate crucial components of central metabolism. The interconnectivity between the synthesis of these specific modifications and their effect on regulatory networks emphasizes the pleiotropic nature of tRNA modifications.

### 3.2. Environmental Stressors

tRNA modifications dynamically respond to changes in the environment including oxidative and hypoxic conditions, ultraviolet radiation (UVR), and extreme growth temperatures. These stressors impact the tRNA epitranscriptome through two main modes: (1) directly, by reacting with modifications already implemented into mature tRNAs, and/or (2) indirectly, by upregulating or inactivating tRNA-modifying enzymes [49,50,51]. In both cases, it is now recognized that these cellular responses modulate tRNA functionality and consequently, the efficiency of translational processes. The expanded chemical functionality of selected tRNA modifications makes these molecules prone to direct interactions with reactive chemical compounds that alter their structural and functional properties. As previously mentioned, some bacterial tRNAs are thiolated at U8 or thiolated/selenated at U34. The location of these thioketone or selenoketone moieties modulate the reactivity of the uridine base and its ability to participate in the thiophilic addition of nucleophiles, Michael addition, oxidation, and cycloaddition reactions (Figure 4) [52,53,54,55,56,57]. In some cases, these characterized reactions have been exploited as experimental tools to improve detection of modified bases [58,59].

#### 3.2.1. Chemical stressors

Thioketone moieties within s-tRNAs are highly reactive towards a variety of alkylating and oxidizing agents. The tautomerization of thioketones allows for facile addition of Michael acceptors (Figure 4Ai). This reaction has been exploited in fluorescent labeling of tRNA to probe dynamic events in translation [60]. In the presence of the alkylating agent iodoacetamide (IAA), a nucleophilic substitution results in the attachment of a carboxyamidomethyl group to C4 of s^4^U (Figure 4Aii). This chemical derivatization amplifies the signal of s^4^U when detected using mass spectrometric methods and metabolic RNA sequencing technologies [57]. Likewise, the reactivity of s^4^U has been explored in the TimeLapse-seq method as it introduces U-to-C mutations into newly synthesized RNA transcripts via treatment of s^4^U with 2,2,2-trifluoroethylamine (TFEA) and meta-chloroperoxybenzoic acid (mCPBA) to form cytidine analogs [61]. Though these reactions were mainly characterized *in vitro*, in vivo analyses demonstrate that the effects of these agents on the tRNA epitranscriptome are not limited to s^2^U and s^4^U modifications. For instance, exposure of *E. coli* and *S. cerevisiae* cells to the alkylating agent methyl methanesulfonate (MMS) resulted in methylation of a percentage of guanosines and adenosines in *E. coli*, and increased levels of m^7^G production in *S. cerevisiae* [51,62].

The oxidative desulfurization of thionucleosides is a well-documented, direct mode of regulation. Earlier studies of *E. coli* tRNA with various chemical reagents showed that hydrogen peroxide, cyanogen bromide, and sodium periodate (NaIO_4_) reacted extensively with the thionucleosides s^4^U, ms^2^i^6^A, 2-thiocytidine (s^2^C), and mnm^5^s^2^u [49]. In the presence of potassium permanganate or periodate, s^4^U is rapidly oxidized to uridine 4-sulfonate. This intermediate is susceptible to participation in further nucleophilic substitution reactions and yields a variety of products, including the formation of cytosine or reversion back to uridine (Figure 4C) [54,55]. The reactivity of s-tRNAs varies with the position of this substitution and its location within the overall tertiary structure of the tRNA. The kinetics of hydrogen peroxide reactions indicate that s^4^U decays three times faster than mnm^5^s^2^u in *E. coli* tRNA, while 2-thioribothymidine (s^2^T54) tRNA decay was 500-fold slower than s^4^U in *T. thermophilus* [63]. Remarkably, the reactivity of these modified bases is distinct when comparing isolated nucleotides with folded and denatured tRNAs, indicating that the overall structure of the tRNA and stacking of s-modified bases control their reactivity against hydrogen peroxide. Sochaka et al. found that under mildly acidic conditions, dethiolation results in the formation of the 4-pyrimidinone (H^2^U) nucleoside (Figure 4Bi), whereas under mildly basic conditions, s^2^U reverts to unmodified U (Figure 4Bii) [53].

Oxidative stress can indirectly modulate the tRNA epitranscriptome by promoting the expression of certain tRNA-modifying enzymes. For instance, in *S. cerevisiae*, hydrogen peroxide exposure yields a significant increase in m^5^C34, 2′-O-methylcytidine (Cm32) and N2,N2-dimethylguanosine (m^2^_2_G26) [51]. The observed increase in m^5^C is linked to the codon-biased translation of UUG-enriched mRNAs, such as the ribosomal protein Rp122a [64]. In the Gram-negative bacterium *Pseudomonas aeruginosa*, hydrogen peroxide improves translation of Phe/Asp enriched mRNAs, such as the catalase genes, *katA* and *katB,* by increasing the TrmB-mediated m^7^G [65]. Likewise, in the phytopathogenic fungus *Colletotrichum lagenarium*, the lack of the *trmB* ortholog, *aph1*, increases the sensitivity of this organism to hydrogen peroxide stress and causes impaired growth, potentially assigning a role for m^7^G in stress tolerance [66]. Furthermore, inactivation of *P. aeruginosa* TrmJ, the enzyme responsible for 2-O’-methylation of adenosine (Am), uridine (Um) and Cm at position 32 of tRNA, decreases expression of catalase genes in pathways involving OxyR regulation [67]. Thus, it is proposed that 2-O’-methylation at position 32 augments protective stress responses during oxidative challenge.

Indirect regulation of the tRNA epitranscriptome is easily achieved through direct reactions of environmental stressors with tRNA-modifying enzymes and their associated cofactors. The effects of oxidative stress are diverse and affect tRNA modifications via distinct routes. For example, tRNA- modifying enzymes that contain Fe-S clusters are direct targets of oxidative stress, leading to inactivation of their metal clusters and consequently, downregulation of their associated modifications. In *Salmonella enterica*, oxidative stress inactivates MiaB, as well as components of Fe-S cluster biosynthesis. These inactivations result in a decrease of ms^2^i^6^A and 2-methylthio-*N*^6^-(cis-hydroxyisopentenyl) adenosine (ms^2^io^6^A), altering translational capacity [50]. Conversely, the Fe-S tRNA-modifying enzyme, TtcA, is associated with the peroxide stress response in *P. aeruginosa.* Inactivation of *ttcA* results in hypersensitivity towards hydrogen peroxide and hypochlorite treatments [68]. TtcA catalyzes the thiolation of C32 on tRNA^Arg1^ (s^2^C32), a rare modification that restricts codon recognition and improves aminoacylation [69]. The [4Fe-4S] cluster of TtcA is critical for its function and highly susceptible to damage by ROS. Interestingly, *ttcA* mRNA expression levels increase 13-fold upon hydrogen peroxide challenge through a transcriptional response involving the OxyR regulator. Moreover, the activity of TtcA is associated with the accumulation of KatA [68]. Although hydrogen peroxide challenge directly inactivates TtcA, it is possible that higher expression levels of this enzyme is a compensatory mechanism. In this model, higher levels of TtcA expression may guarantee enough basal synthesis of s^2^C-modified tRNA to mediate cellular responses against hydrogen peroxide stress.

#### 3.2.2. Respiratory Stressors

Differential accumulation of tRNA modifications is observed in organisms that thrive in both aerobic and anaerobic growth conditions. In response to hypoxia, *Mycobacterium bovis* BCG undergoes tRNA reprogramming and codon-biased translation that results in the expression of stress response proteins. The onset of hypoxia increases uridine 5-oxyacetic acid (cmo^5^U34) found in tRNA^Thr(UGU)^ and increases translation of transcripts enriched in ACG cognate codons, including the master regulator of hypoxia-induced bacteriostasis, DosR [24]. This finding supports the idea of a highly coordinated system of codon-biased mRNAs. Variations in the tRNAome and anticodon modifications are proposed to distinguish synonymous decoding events as mechanisms for regulating the expression of stress response proteins at the translational level [24,70].

In *Salmonella typhimurium*, the hydroxylation of the isopentenyl side chain of ms^2^i^6^A to ms^2^io^6^A is contingent on aerobic growth conditions. A rapid shift from anaerobic to aerobic respiration results in a 5-fold increase in ms^2^io^6^A with a concomitant decrease in ms^2^i^6^A, while leaving the levels of other thionucleosides such as s^2^U, s^4^U, and s^2^C unaffected [71]. The conversion of ms^2^i^6^A to ms^2^io^6^A is catalyzed by the non-heme, diiron, and O_2_-dependent monooxygenase, MiaE [72]. Contrary to expectations, inactivation of *miaE* does not cause delays in growth rate when transitioning from anaerobic to aerobic respiration, ruling out the function of this modification as a sensor of aerobic metabolism. Strikingly, *miaE* knockout strains display defects in central metabolites such as TCA- cycle intermediates, deficiencies of which are associated with the ubiquinone pool. Interestingly, *miaE* and consequently, ms^2^io^6^A, are only present in a subset of facultative anaerobes and are absent in many organisms including *E. coli* and *B. subtilis*, and a substitute for the function of this modification in metabolic response has yet to be identified in these species. Moreover, both *E. coli* and *S. typhimurium* tRNAs accumulate Q under anaerobic growth conditions, while the Q-precursor, epoxyqueuosine (oQ), accumulates during aerobic conditions. Changes in the levels of oQ to Q are attributed to the dependence of oQ-reduction on the oxygen sensitive QueG, an Fe–S/cobalamin-dependent enzyme [73,74,75]. Taken together, despite the range of phenotypes associated with the absence of these modifications, the respiratory-dependent fluctuations in the levels of cmo^5^ U, ms^2^io^6^A, and oQ qualify these modifications as markers of cellular oxygen requirements.

#### 3.2.3. Temperature Stressors

Temperature is an important factor in the stability and reactivity of biomolecules. The adaptation of these molecules in psychrophilic (cold-loving) and thermophilic (heat-loving) organisms is of particular interest. At high temperatures, nucleic acids can undergo denaturation and subsequent thermal degradation [76]. As a result, thermophilic organisms need to structurally stabilize their tRNAs to prevent loss of functionality. One main adaptation is increasing the GC-content of tRNA to raise its melting temperature [77,78]. Other stabilizing factors include the insertion of tRNA modifications, the use of RNA-binding proteins, and the introduction of polyamines [79,80,81]. Post-transcriptional modifications within the ribose and base of tRNA increase the melting temperature and stabilize its structure at the higher end of the thermal spectrum. Simple tRNA modifications such as thiolations, methylations, and even double methylations increase structural rigidity and thermal stability of tRNAs.

In extreme-thermophiles and hyper-thermophiles, double methylated nucleosides such as N2,2′-O-dimethylguanosine (m^2^Gm26), N2,N2,2′-O-trimethylguanosine (m^2^_2_Gm26) and N4-acetyl-2′-O-methylcytidine (ac^4^Cm34) are effective in blocking temperature-induced hydrolytic damage of tRNA. The proposed mechanism for this protective role is attributed to O-methylation, which inactivates the reactive 2′-OH groups, thereby preventing nucleophilic attack onto neighboring phosphodiester bonds. ac^4^Cm prevents temperature-induced hydrolytic cleavage of tRNA by contributing to the formation of a stable helix and promoting a C3′-endo conformation of the adjacent ribose [82,83,84]. Additionally, s^2^T54, a thermophile-specific modified nucleoside, increases the melting temperature of tRNA by 3°C and stabilizes the overall tRNA structure at high temperatures [63,85]. In *T. thermophilus*, increasing cultivation temperature correlates with higher levels of s^2^T and consequently, higher melting temperatures of tRNA [86].

In the hyperthermophile *Thermococcus kodakarensis*, mutant strains lacking archaeosine (G^+^15), N1-methyladenosine (m^1^A58) or 1-methylinosine (m^1^I57) are thermosensitive. The decreased melting temperature of their associated tRNA suggests that these modifications are relevant for growth at elevated temperatures [87]. Modifications such as pseudouridine (ψ) also contribute to higher nucleic acid thermostability. In the anticodon stem loop of *E. coli* tRNA^Lys^, modification of U39 to ψ39 increases the melting temperature by 5 °C and improves base stacking, while strengthening the base pair with A31 [88]. Likewise, an *E. coli* strain lacking the non-essential ψ55 is less resistant to thermal stress, suggesting that modifications to tRNA that improve structural stability at higher temperatures is not a feature exclusive to thermophiles [89].

Though modifications like s^2^T and 2′O-methylations are directly involved in heat resistance, other modifications indirectly impact the level of thermoprotective modifications. For example, in *T. thermophilus*, m^1^A is required for s^2^T incorporation both in vitro and *in vivo*. Inactivation of tRNA m^1^A-methyltransferase, TrmI, leads to undetectable levels of m^1^A and reduces the levels of s^2^T to only 15% of wild type levels. Additionally, the *T. thermophilus ∆trmI* strain is unable to grow at 80 °C and tRNA isolated from this mutant strain displays a lower melting temperature [90]. Another example of indirect effect is m^7^G. Lack of this modification leads to severe growth defects at 80 °C and decreases melting temperatures and half-lives of tRNA as a result of hypomodifications at several positions. This finding suggests that m^7^G may serve as a prerequisite for other modifications harboring functions in heat stress adaptation [91]. Identification of specific nucleosides that contribute to the stability of tRNA can be quite challenging due to the complexity and sequentiality of modification networks.

Similar to extremely hot habitats, cold temperature environments are challenging for organisms due to key barriers such as decreased membrane fluidity, reduced enzyme activity, and metabolism depression [92,93,94]. For psychrophilic organisms, an increase in the structural flexibility of biomolecules is the most common mechanism of adaptation. Although the mechanism underlying how tRNA adjusts to low temperatures is not well understood, the psychrophilic shrimp *Euphausia superba* has a decreased tRNA GC-content that allows for increased structural flexibility [95]. Similarly, post-transcriptional modifications such as dihydrouridine (D) and ψ play important roles during psychrophilic adaptation. Dihydrouridine is the only tRNA modification that cannot stack with other bases and employs the C2′- endo sugar pucker usually observed at lower temperatures. As a result, tRNA regions enriched in D are more flexible [96,97,98,99]. Likewise, ψ also contributes to the flexibility and functionality of tRNA at low temperatures. Lack of ψ55 in *T. thermophilus* results in a growth retardation at lower cultivation temperatures, supporting the idea that its absence enhances the incorporation of modifications conferring structural rigidity and thermostability [100].

The tRNA-modifying enzyme TrmE plays a significant role in cold adaptation in the psychrophile *Pseudomonas syringae* [101]. Inactivation of *trmE* in *P. syringae* leads to growth defects in cells cultured at 4 °C, but not at higher temperatures (22 °C). In this organism, a decrease in culture temperature increases the expression levels of *trmE*, suggesting possible transcriptional control by a cold-inducible regulator. The *E. coli* TrmE ortholog, MnmE, is a GTPase involved in the biosynthesis of mnm^5^ s^2^U tRNA [102]. The GTPase activity of *P. syringae* TrmE has recently been demonstrated to be temperature dependent, with the enzyme becoming inactive at 30 °C [103]. The tRNA modifying role of TrmE during cold stress, although expected, remains to be experimentally validated.

#### 3.2.4. UV Radiation Stressors

The effects of UVR have long been known to affect translational processes leading to growth arrest in bacteria. The energy emitted by UVC and UVB radiation damages cells by inducing ssDNA breaks and cyclobutane dimers [104,105]. UVR-induced alterations cause oxidative damage to essential biomolecules in *Pseudomonas sp*. and *Acinetobacter* spp., including proteins and nucleic acids [106]. In *E. coli*, moderate levels of UVA exposure results in a precipitous decrease in s-tRNA modifications such as s^2^C, s^4^U, mnm^5^s^2^U, carboxymethylaminomethyl-2-thiouridine (cmnm^5^s^2^U34), and ms^2^i^6^A. UVR directly destabilizes the tRNA tertiary structure, leading to secondary oxidation reactions of otherwise exposed bases. Accumulation of 8-oxoguanosine (8-oxo-rG) is proposed to be an indirect side effect of UVR exposure and is associated with the loss of mnm^5^s^2^U. An *E. coli* mutant strain lacking the mnm^5^s^2^U-modifying enzyme, TrmU, shows markedly higher levels of 8-oxo-rG when compared to the wild-type. This finding proposes that mnm^5^s^2^U exerts protective effects on tRNAs when faced with harmful oxidizing agents, since this modification is known to stably wobble base with G-ending codons [107].

In response to the demand imposed by harmful UVR, various *Bacillus spp.* such as *B. subtilis*, *B. anthracis* and *B. cereus* have developed protective mechanisms allowing them to form spores that are 10 to 50 times more resistant to this condition [108,109]. In *B. subtilis*, the synthesis of s^4^U-tRNA involves the synchronous efforts of dedicated enzymes [110]. It has been long documented that exposure of s^4^U-modified tRNA to UVR forms a covalent crosslink adduct with the neighboring cytosine 13 (Figure 4D). tRNAs containing these adducts display altered conformations and induce the stringent response [52]. In *E. coli*, UV-illuminated, s^4^U-photocrosslinked tRNA^Phe^ and tRNA^Pro^ impair the ability of these tRNAs to be charged by their corresponding aaRSs by more than 25-fold [20]. These uncharged tRNAs are then ushered to the A-site of the ribosome where protein synthesis stalls and a growth delay is observed [111,112]. This phenotype is exacerbated due to the fact that ribosome stalling also triggers the synthesis of ppGpp, abrogating the production of stable RNA and yielding a growth lag [113]. In *S. typhimurium*, UV-induced damage to s^4^U is proposed to elicit this response via two routes; (i) inducing the synthesis of the *relA*-dependent stringency factor, ppGpp and, (ii) increasing dinucleotide production as a result of perturbed s^4^U interactions with specific aaRSs, with concomitant ApppGpp accumulation. Increased levels of ppGpp and ApppGpp then act as secondary messengers for the expression of specific proteins known to increase cell survival [111]. The reactivity of s^4^U-modified tRNA provides a clear example of tRNA modifications serving as direct sensors of UV radiation that contribute to bacterial protective cellular responses.

## 4. Pathogenicity, Virulence and Antibiotic Targeting

The pivotal role of modified tRNA species in microbial translation makes these molecules attractive targets for the development of antibiotics. While many tRNA modifications and their corresponding biosynthetic enzymes are universally conserved in all three domains of life, others are conserved only among bacteria. Despite the ubiquitous nature of these modifications, very few are actually essential in bacteria [114], effectively limiting the range of biosynthetic enzymes that can be used as antibiotic targets. The tRNA methyltransferase D (TrmD) is under investigation as one of these possible targets [115]. TrmD catalyzes the methylation of guanine at position 37 of some tRNAs to form 1-methylguanosine (m^1^G), a modification that prevents frameshifting at the A-site of the ribosome [116,117,118]. TrmD orthologs are essential in various human pathogens such as *Salmonella enterica* serovar *typhimurium* (hereafter *Salmonella*)*, P. aeruginosa*, *Streptococcus pneumoniae*, and *E. coli*, as well as non-pathogenic model organisms, like *B. subtilis* [115,119,120,121]. In *E. coli* and *Salmonella*, a decrease in m^1^G levels impairs the translation of membrane proteins involved in bacterial multidrug resistance and thus, sensitizes these bacteria to multiple classes of antibiotics [122]. With the increasing resistance of *P. aeruginosa* to antibiotics, a recent advance in determining the crystal structure and catalytic mechanism of *P. aeruginosa* TrmD represents a step in the right direction. A complete understanding of the unique reaction mechanisms these enzymes employ is necessary for upstream application using tRNA-modification machinery as antimicrobial therapeutic targets [117].

The identification of conserved tRNA-modifying enzymes in various pathogens provides a platform for many groups to test the efficacy of this model and to catalog the associated phenotypic effects on strain physiology. The foodborne pathogen, *S. enterica,* uses GidA/MnmE to create the fully modified mnm^5^s^2^U [123]. In a GidA/MnmE null background, *S. enterica’s* ability to infiltrate intestinal epithelial cells is reduced by 1000-fold, drastically affecting pathogenicity [124]. Gram-negative pathogens, such as *Shigella flexneri,* require not one, but two post-transcriptional modifications for adequate expression of their virulence factor proteins. In this case, ms^2^i^6^A is required for the expression of the *virF* gene, a virulence transcription factor regulator implicated in *S. flexneri* pathogenesis. The absence of this modification results in poor translation of *virF* mRNA and reduces expression of other virulence genes [125]. These findings are concurrent with a characteristic role of modifications at position A37 and their role in reading frame maintenance and translational efficiency. In addition to the ms^2^i^6^A modification, the elaborate Q modification was later shown to have a substantial impact on *S. flexneri* cell growth and pathogenicity. Hence, the absence of both of these modifications substantially reduces virulence and *S. flexneri* replicative competency [126]. Taken together, these results imply that the absence of mnm^5^ s^2^U, ms^2^i^6^A alone, or in tandem with Q, have a deleterious effect on non-typhi *Salmonella* and *S. flexneri* virulence, making these modifications appealing candidates for the clinical interruption of microbial pathogenesis.

Pathogen-linked tRNA modifications have not only been investigated for their potential antimicrobial properties, but the reduced virulence phenotypes associated with hypomodification holds great potential in live-attenuated vaccine development. This method relies on the creation of attenuated live vaccines from populations of pathogenic bacteria that possess genetically engineered reduced virulence genotypes. These genetic variants can then astimulate an adequate humoral response from the host without the side effects of an established bacterial infection. This model is being explored in the Gram-positive pathogen *Streptococcus pyogenes*, the etiological agent of diseases such as necrotizing fasciitis, scarlet fever, and many more alike. Infectious strains of *S. pyogenes* lacking the aforementioned GidA/MnmE mnm^5^s^2^U-modifying enzymes exhibited a global reduction in the expression of certain virulence factors, while also being able to elicit a host immune response [127]. The same strategy is being employed in *S. enterica* using a GidA deficient strain to produce lower virulence strains for prophylactic host treatment and protection [128]. These new methods of exploiting bacterial tRNA modifications as a route of virulence control may provide a potential link between the realms of basic scientific research and clinical application.

## 5. Discussion and Concluding Remarks

Modified tRNAs have classically been regarded as the unchanging mediators of translation, as well as determinants for efficient and accurate aminoacylation. However, mounting evidence allows for reexamination of their potential roles as sensors, markers, or regulators of cellular status. Many modification levels fluctuate based on nutrient availability, growth temperature, oxidative stress, and exposure to UV light (Figure 3). These changes are a result of direct interactions with specific stressors or the indirect outcomes of altered regulation of biosynthetic enzymes at the transcriptional and post-translational levels.

Dynamic changes in the tRNA epitranscriptome grant significant regulatory potential to modified nucleotides in cellular stress responses and translational reprogramming. The idea that modified tRNAs assume regulatory roles provides a fascinating model that defies previously established canon concerning the use of tRNA exclusively during translation. These assumptions allow us to envision a scenario where these chemical moieties take a front seat role in dictating the flux of metabolism in response to external stimuli. These assumptions are highlighted in the *E. coli* regulatory model where ms^2^i^6^A levels are coupled with iron homeostasis through a series of sequential and interconnected processes, presumably instigated and mediated by ms^2^i^6^A production. The direct reactivity of tRNA modifications with specific stressors suggests that they can be used as biomarkers for exposure. In yeast, hydrogen peroxide exposure consistently increases the levels of m^5^C, Cm, and m^2^_2_G, while MMS exposure exerts the same effects on the levels of m^7^G. While these accumulating modified species do not appear to be an integral part of any sensory or response pathway, these modification profiles may serve as markers of global metabolic status.

Understanding the importance of tRNA modifications in mediating stress responses and self- defense pathways affords a basis for broad use in basic science and clinical application. Based on growing evidence, targeting tRNA-modifying pathways in human pathogens offers a strategy to combat microbial infection. While tRNA modifications clearly have a more sizable role in regulating cell growth and participating in molecular signaling, they are still one of many intersecting and complex processes that must be taken into consideration when making generalized statements about the cell’s response to suboptimal metabolic events. Ongoing investigations will continue to uncover the mechanisms by which modifications are able to respond to cellular stressors and potentially tune metabolic processes at the epigenetic and translational levels.

## Figures and Tables

**Figure 1 genes-11-00907-f001:**
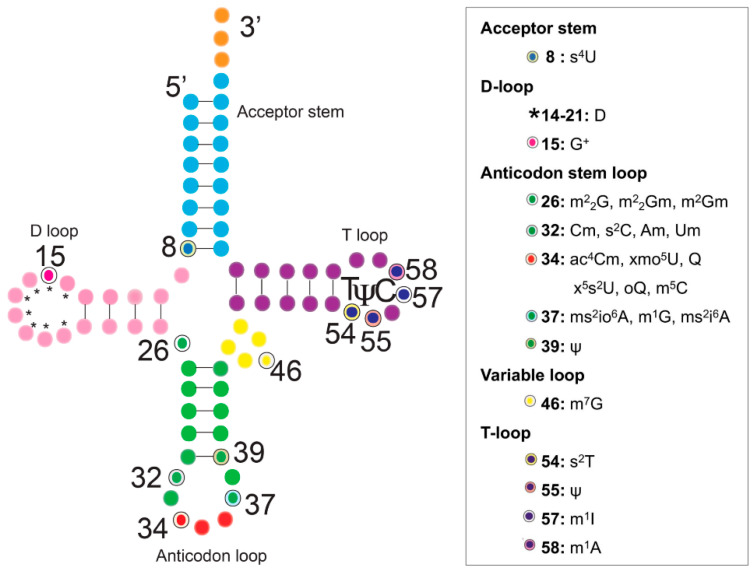
Schematic of the tRNA secondary structure and location of modified bases. Each structural domain of tRNA is represented in a different color and modified bases known to be affected by environmental and/or nutritional conditions are highlighted.

**Figure 2 genes-11-00907-f002:**
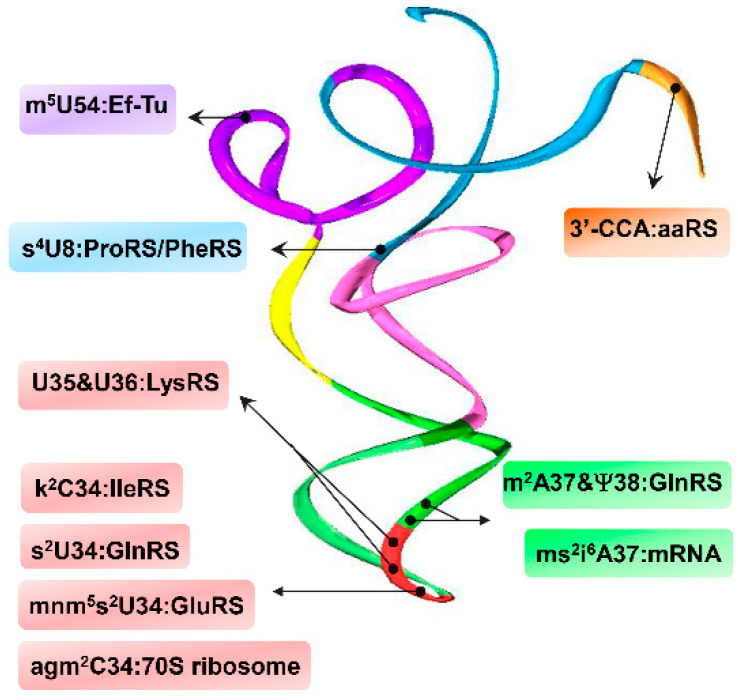
Modifications throughout the tRNA structure are important recognition elements for interaction with a variety of partners in translation. The tRNA structure (PDB 1u0b) shows domains color coded in accordance with Figure 1. m^5^U54 confers binding with *T. aquaticus* Ef-Tu [19]; s^4^U8 is necessary for structural stability impacting interactions with *E. coli* PheRS and ProRS [20]; U35/U36, along with mnm^5^s^2^U34, are major identity elements for *E. coli* LysRS binding [17]; agm^2^C34 interacts with the *T. thermophilus* 70S ribosome [15]; k^2^C34 is required for codon discrimination by IleRS in *E. coli* and *B. subtilis* [14]; s^2^U34, m^2^A37, and ψ38 are required for *E. coli* GlnRS substrate recognition and aminoacylation efficiency [16,18]; ms^2^i^6^A37 directly interacts with cognate mRNA bases in E, P and A sites of the ribosome [8]; all aaRSs require 3′-CCA moiety for tRNA charging [7].

**Figure 3 genes-11-00907-f003:**
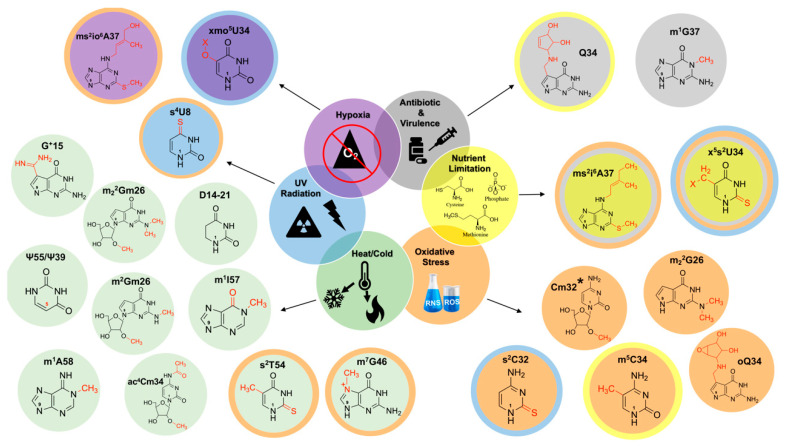
Schematic representation of tRNA modifications affected by nutritional and environmental stressors. The description of these factors, their associated modifications along with references are also noted in Appendix A. Modifications affected by more than one condition are indicated with corresponding colored rings. *2-O’-methyladenosine (Am32) and 2-O’-methyluridine (Um32) are additionally affected by oxidative stress. The numbers “1” and “9” found within each base denote the β-glycosidic bond orientation found in the pyrimidine and purine bases, with their adjacent riboses in tRNA. The structure of the full nucleoside is only shown for modifications containing 2′-O-methylation.

**Figure 4 genes-11-00907-f004:**
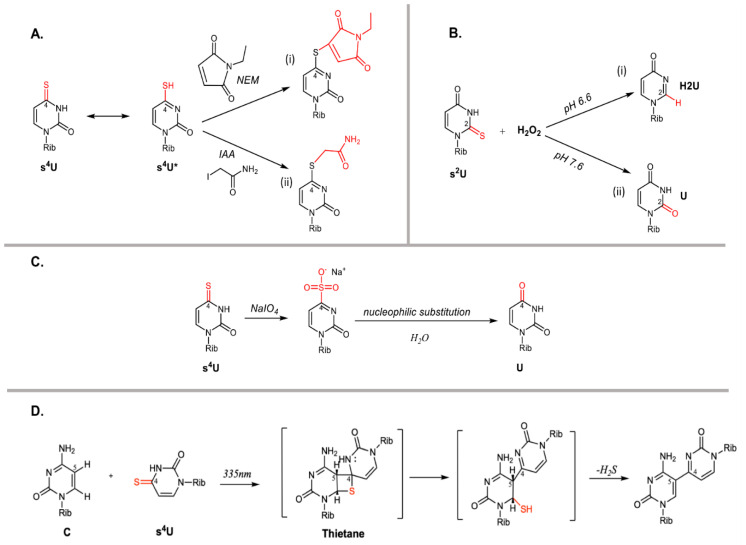
Direct reactions involving thiouridines. (**A**) Alkylation reactions of reactive thiols are shown through (i) thiol-Michael addition of s^4^U with N-methylmaleimide (NEM) [56] and (ii) nucleophilic substitution with iodoacetamide (IAA) [57]; (**B**) Oxidation of s^2^U with hydrogen peroxide yields the formation of (i) 4-pyrimidinone under mildly acidic conditions and (ii) uridine under mildly basic conditions [53]; (**C**) Oxidation of s^4^U with sodium periodate (NaIO_4_) leads to the formation of U [49]; (**D**) UV radiation promotes the radical-catalyzed cycloaddition reaction of s^4^U and C [52].

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
