# Peer review of "Extracurricular Functions of tRNA Modifications in Microorganisms"

_genes, 2020, doi:10.3390/genes11080907_

Round 1

Reviewer 1 Report

Edwards et al. review how cellular conditions in microorganisms affect the modifications in tRNA. The concept is suitable for one of the reviews in Genes. Before the publication, the authors need to improve the readability.  

Figure 1

Several major modifications focused in this review should be indicated at their corresponding positions in the tRNA structure: for example, ms2i6A37 and m1G37 for the position 37.

Figure 3

The chemical structural formulae are too small to read. In addition, the formulae of all the modifications mentioned in this review should be indicated in this figure or in any figure.

Related to Figures 1 and 3, it is helpful if the authors present any table showing “position in tRNA”, “modification”, “(estimated) function”, “effect by cellular conditions”, and “reference”, for easier understanding.

Minor point

Line 62: Typo, re -> are

Author Response

We thank both reviewers for careful analysis of the manuscript content and writing. We have revised the document to best address both reviewer comments which includes changes to the figure 1, 2, and 3 and substantial changes to the text. Please see our point by point response in italics text.

Edwards et al. review how cellular conditions in microorganisms affect the modifications in tRNA. The concept is suitable for one of the reviews in Genes. Before the publication, the authors need to improve the readability.  

The manuscript text has been substantially revised to improve readability and to address the other reviewer’s comments.

Figure 1 - Several major modifications focused in this review should be indicated at their corresponding positions in the tRNA structure: for example, ms2i6A37 and m1G37 for the position 37.

Figure 1 has now a key legend indicating the modifications in each position.

Figure 3 - The chemical structural formulae are too small to read. In addition, the formulae of all the modifications mentioned in this review should be indicated in this figure or in any figure.

For modifications within the base, we have removed the ribose to give space to increase the sizes of these structures.

Figure 3 has been redone with the structure of modified bases (instead of nucleotides). Related to Figures 1 and 3, it is helpful if the authors present any table showing “position in tRNA”, “modification”, “(estimated) function”, “effect by cellular conditions”, and “reference”, for easier understanding.

A Supplemental table has now been added that provides the information about the location, modification, cellular conditions affecting the levels of these modifications along with accompanying references.

We thank the reviewer for constructive feedback. We hope to incorporated the reviewer’s suggestions into the revised document.

Reviewer 2 Report

The content of this review could make a nice contribution to the field. The authors cover a timely and often overlooked area that allows cells to respond to the environment. Some sections are slightly under-referenced but overall many key examples with appropriate references are noted. A similar review of narrower scope was published by Gupta and Laxman in Current Genetics in May and there are many other reviews on tRNA modification, but I don’t think that these preclude publication of this work.

Unfortunately this work is not in a form suitable for publication because of the writing. It is a very tedious read. It is far too long not because of the content but because of the writing. I estimate that with a little effort that the manuscript could be 30+% shorter without changing the content by making the writing more concise and removing redundancies. Below are some example sentences that I have edited to be more concise (character counts of before and after follow each). This type of editing should be done throughout the manuscript. Avoid unnecessary adjectives and adverbs.  Have been shown to be = is, avoid is able to, etc. Avoid nominalizations. Let the verbs do the taking. For example: display accumulations becomes, accumulates; have been reported to differentially accumulate becomes differentially accumulate; that are capable of thriving becomes thrive; leads to a rapid increase becomes increases. Eliminate unnecessary Furthermore, In addition, etc. Where possible phrase in the positive rather than the negative. Avoid contradictory arguments and qualifiers unless they are necessary (e.g line 154).

------------

Furthermore, genes encoding proteins that have shown to be involved in glucose and carbohydrate metabolism are found to be enriched in codons that are preferentially read by thiolated tRNAs.  190

Furthermore, genes encoding proteins involved in carbohydrate metabolism are enriched in codons preferentially read by thiolated tRNAs.  136

In addition to responses elicited during fluctuations of central metabolites, iron availability significantly impacts the degree of modification to A37. The E. coli tRNA-modifying enzyme, MiaB, implements the ms2i6A modification and is dependent on two types of associated Fe-S clusters for its activity, making it highly sensitive to intracellular iron metabolism and oxidative stress.  333

Iron availability also impacts modification of A37. Two types of Fe-S clusters make the E. coli ms2i6A tRNA-modifying enzyme, MiaB, highly sensitive to intracellular iron metabolism and oxidative stress.  175

Additionally, in the Gram-negative bacterium Pseudomonas aeruginosa, exposure to hydrogen peroxide leads to increased levels of TrmB-mediated m7G, a modification of which improves translation of Phe/Asp enriched mRNAs such as the catalase genes, katA and katB [67].  229

In Pseudomonas aeruginosa, a Gram-negative bacterium, hydrogen peroxide improves translation of Phe/Asp enriched mRNAs such asthe catalase genes, katA and katB by increasing TrmB-mediated m7G modification [67]. 184

Adaptive cellular mechanisms that have arisen in response to a wide range of environmental temperatures include changes in the overall make up of tRNA modifications. 141

Adaptive cellular mechanisms alter tRNA modifications in response to temperature change. 78

UVC and UVB types of radiation are widely regarded as the most damaging to living cells because the energy emitted at these wavelengths is sufficient to induce the formation of ssDNA breaks and cyclobutane dimers 178

The energy emitted by UVC and UVB radiation damages cells by inducing ssDNA breaks and cyclobutene dimers 89

-------------------

Word usage is sometimes inappropriate. And remember that English is not the first language for all readers so use or rarely used words may mean a google search. If in doubt keep it simple.

An example of a redundancy is:

The E. coli tRNA-modifying enzyme, MiaB, 270 implements the ms2i6A modification and is dependent on two types of associated Fe-S clusters for its 271 activity, making it highly sensitive to intracellular iron metabolism and oxidative stress [48].

Concept is repeated essentially in:

In Salmonella enterica, oxidative stress inactivates the Fe-S cluster-dependent tRNA-360 modifying enzyme, MiaB, as well as components of Fe-S cluster biosynthesis. These inactivations 361 result in a decrease of ms2i6A and 2-methylthio-N6-(cis-hydroxyisopentenyl) adenosine (ms2io6A) 362 modifications, ultimately altering translational capacity [52].

Other points.

The theme of the paragraph beginning on line 62 is confusing.

Refer to the investigators that performed the work. This does not need to be done constantly but it makes the writing more active.

Line 74: What is the common loop.

Figure 2 is a little awkward because it shows interaction with an aaRS but documents interactions with other factors.

Be clear what the modification is or refer specifically to the modified base. The modification is not the modified base. (e.g. line 248).

There are other examples were the writing is unclear but they should be fixed when a thorough editing is performed.

Author Response

We thank both reviewers for careful analysis of the manuscript content and writing. We have revised the document to best address both reviewer comments which includes changes to the figure 1, 2, and 3 and substantial changes to the text. Please see our point by point response in italics.

The content of this review could make a nice contribution to the field. The authors cover a timely and often overlooked area that allows cells to respond to the environment.

Some sections are slightly under-referenced but overall many key examples with appropriate references are noted. A similar review of narrower scope was published by Gupta and Laxman in Current Genetics in May and there are many other reviews on tRNA modification, but I don’t think that these preclude publication of this work.

The manuscript now includes 128 references that were deemed relevant to the topic discussed in this review. The original work of Laxman and collaborators is referenced in this manuscript and provides important contributions to the understanding of s2U34 in metabolic processes. While their 2019 review in Current Genetics focused on s2U34 and sulfur availability, we believe our manuscript provides a broader survey of the literature discussing the impact of additional environmental factors in tRNA modifications.

Unfortunately this work is not in a form suitable for publication because of the writing. It is a very tedious read. It is far too long not because of the content but because of the writing. I estimate that with a little effort that the manuscript could be 30+% shorter without changing the content by making the writing more concise and removing redundancies.

Below are some example sentences that I have edited to be more concise (character counts of before and after follow each). This type of editing should be done throughout the manuscript. Avoid unnecessary adjectives and adverbs.  Have been shown to be = is, avoid is able to, etc. Avoid nominalizations. Let the verbs do the taking. For example: display accumulations becomes, accumulates; have been reported to differentially accumulate becomes differentially accumulate; that are capable of thriving becomes thrive; leads to a rapid increase becomes increases. Eliminate unnecessary Furthermore, In addition, etc. Where possible phrase in the positive rather than the negative. Avoid contradictory arguments and qualifiers unless they are necessary (e.g line 154).

We have made an honest attempt to address the reviewer’s concern about the writing. We conducted substantial editing through the manuscript and removed over 1,000 words from the document (~15% of the main text). We provide in this resubmission the modified document using track changes, as well as a revised copy of the document. These stylistic changes were intentional to make the review narrative more active, without compromising the accuracy of the information presented.

Furthermore, genes encoding proteins that have shown to be involved in glucose and carbohydrate metabolism are found to be enriched in codons that are preferentially read by thiolated tRNAs.  190

Furthermore, genes encoding proteins involved in carbohydrate metabolism are enriched in codons preferentially read by thiolated tRNAs.  136

This statement was modified as suggested.

In addition to responses elicited during fluctuations of central metabolites, iron availability significantly impacts the degree of modification to A37. The E. coli tRNA-modifying enzyme, MiaB, implements the ms2i6A modification and is dependent on two types of associated Fe-S clusters for its activity, making it highly sensitive to intracellular iron metabolism and oxidative stress.  333

Iron availability also impacts modification of A37. Two types of Fe-S clusters make the E. coli ms2i6A tRNA-modifying enzyme, MiaB, highly sensitive to intracellular iron metabolism and oxidative stress.  175

This statement was modified as suggested.

Additionally, in the Gram-negative bacterium Pseudomonas aeruginosa, exposure to hydrogen peroxide leads to increased levels of TrmB-mediated m7G, a modification of which improves translation of Phe/Asp enriched mRNAs such as the catalase genes, katA and katB [67].  229

In Pseudomonas aeruginosa, a Gram-negative bacterium, hydrogen peroxide improves translation of Phe/Asp enriched mRNAs such as the catalase genes, katA and katB by increasing TrmB-mediated m7G modification [67]. 184

This statement was modified as suggested.

Adaptive cellular mechanisms that have arisen in response to a wide range of environmental temperatures include changes in the overall make up of tRNA modifications. 141

Adaptive cellular mechanisms alter tRNA modifications in response to temperature change. 78

This statement was modified as suggested.

UVC and UVB types of radiation are widely regarded as the most damaging to living cells because the energy emitted at these wavelengths is sufficient to induce the formation of ssDNA breaks and cyclobutane dimers 178

The energy emitted by UVC and UVB radiation damages cells by inducing ssDNA breaks and cyclobutene dimers 89

This statement was modified as suggested.

Word usage is sometimes inappropriate. And remember that English is not the first language for all readers so use or rarely used words may mean a google search. If in doubt keep it simple.

The text has been revised and word usage addressed in several instances.

An example of a redundancy is: The E. coli tRNA-modifying enzyme, MiaB, 270 implements the ms2i6A modification and is dependent on two types of associated Fe-S clusters for its 271 activity, making it highly sensitive to intracellular iron metabolism and oxidative stress [48].

Concept is repeated essentially in:

In Salmonella enterica, oxidative stress inactivates the Fe-S cluster-dependent tRNA-360 modifying enzyme, MiaB, as well as components of Fe-S cluster biosynthesis. These inactivations 361 result in a decrease of ms2i6A and 2-methylthio-N6-(cis-hydroxyisopentenyl) adenosine (ms2io6A) 362 modifications, ultimately altering translational capacity [52].

We understand the reviewer’s comment and we have addressed these redundancies when possible. The fact that certain modifications are affected by more than one condition or factor makes it challenging to provide critical information only once in the text. Nevertheless, we have abbreviated additional information in these cases.

Other points.

The theme of the paragraph beginning on line 62 is confusing.

This paragraph, now starting on line 52 has been restructured. Thanks for point this out.

Refer to the investigators that performed the work. This does not need to be done constantly but it makes the writing more active.

We have added references to the work of some of the investigators as suggested.

Line 74: What is the common loop.

The sentence including this statement was removed.

Figure 2 is a little awkward because it shows interaction with an aaRS but documents interactions with other factors.

We have modified figure 2 to only display the tRNA structure.

Be clear what the modification is or refer specifically to the modified base. The modification is not the modified base. (e.g. line 248).

This was rectified several places in the document.

There are other examples were the writing is unclear but they should be fixed when a thorough editing is performed.

We thank the reviewer for careful evaluation of the writing in this review. We hope to have addressed the concerns raised in the initial review period.